# Evaluation of plasma anti-GPL-core IgA and IgG for diagnosis of disseminated non-tuberculous mycobacteria infection

**Arnone Nithichanon**[1,2☯], **Waraporn Samer**[1☯], **Ploenchan Chetchotisakd**[2], **Chidchamai Kewcharoenwong**[1,3], **Manabu Ato**[4], **Ganjana Lertmemongkolchai**[1,3]*

1 Cellular and Molecular Immunology Unit, Centre for Research and Development of Medical Diagnostic Laboratories, Faculty of Associated Medical Sciences, Khon Kaen University, Khon Kaen, Thailand, 2 Division of Infectious Diseases and Tropical Medicine, Department of Medicine, Faculty of Medicine, Khon Kaen University, Khon Kaen, Thailand, 3 Faculty of Associated Medical Sciences, Chiang Mai University, Chiang Mai, Thailand, 4 Department of Mycobacteriology, National Institute of Infectious Diseases, Tokyo, Japan

☯ These authors contributed equally to this work.
* ganja_le@kku.ac.th

**Data Availability Statement:** All relevant data are within the manuscript and its Supporting Information files.

## Abstract

Detection of IgA antibody against *Mycobacterium avium* complex (MAC) glycopeptidolipid (GPL) has recently been shown to improve the diagnosis of MAC pulmonary disease but has yet to be tested in disseminated Non-tuberculous mycobacteria (NTM) infection. In this study, we address the diagnostic efficacies of an anti-GPL-core ELISA kit in disseminated lymphadenopathy patients positive for NTM culture and anti-IFN-γ autoantibodies. The study was conducted in a tertiary referral center in northeastern Thailand and patients with NTM, tuberculosis, melioidosis, and control subjects were enrolled. Plasma immunoglobulin A (IgA) and G (IgG) antibodies against GPL-core were detected in the subjects and the specificity and sensitivity of the assay was assessed. Anti-GPL-core IgA and IgG levels were significantly higher in NTM patients than other groups (p < 0.0001). Diagnostic efficacy for NTM patients using anti-GPL-core IgA cut-off value of 0.352 U/ml showed good sensitivity (91.18%) and intermediate specificity (70.15%). Using a cut-off value of 4.140 AU/ml for anti-GPL-core IgG showed the same sensitivity (91.18%) with increased specificity (89.55%) and an 81.58% positive predictive value. Most patients with moderate levels (4.140–7.955 AU/ml) of anti-GPL-core IgG had rapidly growing mycobacteria (RGM) infection. Taken together, the detection of anti-GPL-core antibodies could provide a novel option for the diagnosis and management of disseminated NTM infected patients.

## Introduction

Non-tuberculous mycobacteria (NTM) are poorly pathogenic for humans but they can cause infection in people with underlying conditions [1]. The prevalence of opportunistic infections with NTM in pre-disposed hosts has been increasing worldwide [2–5]. Infection with NTM

**Funding:** This study is partly supported by the Japan Agency for Medical Research and Development (AMED) (grant number: JP20fk0108129, JP20fk0108139) and the Atlantic Philanthropies Director/Employee Designated Gift Program. Arnone Nithichanon was supported by the Post-Doctoral Training Program from KKU Research Affairs and the Graduate School, Khon Kaen University (Grant No. 59260).

**Competing interests:** The authors have declared that no competing interests exist.

can be localized or disseminated and have different clinical manifestations [6]. Disseminated NTM infection in adults with autoantibodies against interferon-γ (IFN-γ) has been reported in Southeast Asia and the United States, and is known as anti-IFN-γ autoantibody-associated immunodeficiency syndrome [7]. Lymphadenopathy is the most common clinical outcome observed in Thai patients with anti-IFN-γ autoantibodies, and rapidly growing mycobacteria (RGM), especially *Mycobacterium abscessus*, are the most common cause of disseminated infection [7–10].

Diagnosis of NTM infection by bacterial culture or immunofluorescence lacks sensitivity and is time consuming [11]. It is also an invasive procedure that needs to be performed by an expert clinician for patients with lymphadenopathy [12]. To date, diagnosis of disseminated NTM infection in non-immunocompromised patients by detection of anti-IFN-γ autoantibody has been recommended [13]. Although detection of autoantibodies is useful for clinicians to manage cases, indicators of active NTM infection in patients are still necessary for the follow-up of patients on long-term multidrug therapy [14]. Serological assays for detection of antibodies against NTM's glycopeptidolipid (GPL)-core have been developed for diagnosis of pulmonary NTM infection, especially infection by *M. avium* complex (MAC) [15–17]. The GPL-core is composed of fatty acids, three amino acids and rhamnose [18], which is a major antigen present on the cell wall of several *Mycobacterium* species, excluding *M. tuberculosis* (MTB) complex and *M. kansasii* [17]. According to previous reports, anti-GPL-core immunoglobulin A (IgA) antibody is the most powerful antibody subclass for diagnosis of pulmonary MAC infection [19], with satisfactory diagnostic efficacies in healthy, MTB, or other pulmonary infections, however, the ability to discriminate MAC from non-MAC NTM is still difficult due to cross-reactivity among NTM species [20]. Even though the anti-GPL-core IgA kit was developed for pulmonary MAC infection diagnosis, application of the kit for diagnosis of disseminated NTM infection in patients with lymphadenopathies may be possible.

In this study, surplus plasma samples from Thai patients suspected of disseminated NTM were tested with an anti-GPL-core IgA kit and the results were analyzed along with their laboratory and medical records. Since our cases were disseminated NTM infection with lymphadenopathies, we modified the kit to also detect IgG to compare the diagnostic efficacies of IgG and IgA.

## Materials and methods

### Sample collection and definitions

A total of 137 surplus heparinized plasma samples from the routine service of anti-IFN-γ antibody detection at Srinagarind Hospital, Khon Kaen, Thailand during 2015 to 2018 were recruited along with patient's laboratory and medical information under ethics approval from Khon Kaen University Ethics Committee in human research (HE612234). Collection of heparinized blood samples from adult (aged ≥ 18 years) patients who had *M. tuberculosis* pulmonary infection (MTB, n = 18), *Burkholderia pseudomallei* infection (BP, n = 19), and adult (aged ≥ 18 years) healthy controls (HC, n = 30) were given ethical permission for human research by Nakhon Panom Hospital Ethics committee (NKP$_1$-No.58/2559). Written informed consent was obtained from all participants and no minor was recruited in this study.

Suspected patients with NTM infection were defined according to the criteria set out in our previous studies [21, 22]; (a) patients with lymphadenitis, or (b) signs of infection at any site with reactive skin disease (i.e., Sweet's syndrome, pustular psoriasis, or erythema nodosum), or (c) disseminated infection (infection in more than 1 organ), or (d) co-infection with another opportunistic pathogen (i.e., *Cryptococcus neoformans*, non-typhoidal *Salmonella*,

*Histoplasma capsulatum*, *Talaromyces marneffei*). NTM species were identified by PCR-based assay at Srinagarind Hospital.

Non-immune deficiency cases with pulmonary *M. tuberculosis* infection control were patients previously diagnosed by a clinician and were acid fast stain (AFB) positive from sputum [23] the day before blood sample collection. Cases with *B. pseudomallei* infection were patients with hemoculture positively confirmed [24] the day before blood sample collection. Healthy controls were enrolled according to blood donation guideline of Blood Bank, Nakhon Panom Hospital

## Measurement of anti-GPL core IgA and IgG antibody from human plasma samples

Concentration of plasma anti-GPL core IgA antibody was measured by using GPL core IgA ELISA kit following the manufacturer's instruction (Capilia MAC Ab ELISA, Cat. CAMC8170, Tauns Laboratory Inc., Shizuoka, Japan). Optical density (O.D.) was analyzed with standard curve and results are reported as U/ml.

To detect anti-GPL-core IgG, the antigen adsorbed plate of the IgA kit was used to perform the modified test. Briefly, plasma samples were diluted and added as instructed by the manufacturer. The plate was then incubated at room temperature for 1 hour before washing 4 times with wash solution. Biotinylated mouse anti-human IgG (Clone G18-145, Cat. 555785, BD Biosciences, USA) and HRP-conjugated strepavidin (Cat. 554066, BD Biosciences, USA) were applied instead of the IgA detection antibody from the GPL core IgA ELISA kit, and incubated at room temperature for 1 hour. After washing, chromogen solution was added and the plates were left for the color to appear before stopping the reaction with 2 N $H_2SO_4$. Optical density was measured at 450 nm by ELISA reader (Tecan Magellan, Switzerland). Results were analysed with in-house standard curve selected from a reference sample, diluted to the concentration at 10, 7.5, 5, 2.5 and 1.25 Arbitrary unit/ml (AU/ml), the unknow samples were interpolated from the standard curve by linear regression using GraphPad Prism Software (S1 Fig). Any samples with anti-GPL-core IgG more than 10 AU/ml were diluted and repeated the assay.

## Anti-IFN-γ autoantibody inhibition titer

Anti-IFN-γ autoantibody inhibition titer was determined by inhibitory ELISA as previously described elsewhere [22]. Briefly, 1:250 diluted human IFN-γ capture antibody (Cat. 555142, BD Biosciences, USA) in bicarbonate buffer (pH 9.6) was coated onto a 96 well polystyrene plate (Nunc, Denmark) and incubated overnight at 4°C. The plate was washed with 0.05% Tween-20 in phosphate buffer saline (PBS), then non-specific binding was blocked with 10% fetal bovine serum (FBS) in PBS (pH 7.0) at room temperature for 2 hours. Plasma samples were pre-incubated with 300 pg/ml recombinant human IFN-γ (Cat. 554617, BD Biosciences, USA) on another plate, the final plasma dilutions were at 1:100, 1:1,000, 1:5,000, 1:10,000, 1:50,000, 1:100,000, 1:200,000 and 1:400,000. The pre-incubation plasma plate was left at 37 °C for 1 hour to allow autoantibodies to bind to recombinant human IFN-γ. The capture antibody coated plate was washed and samples from the pre-incubation plasma plate were added before incubating at room temperature for 1 hour to allow unbound recombinant human IFN-γ to bind capture antibody. The plate was washed and 1:250 diluted biotinylated mouse anti-human IFN-γ (Cat. 555142, BD Biosciences, USA) and 1:250 diluted horseradish peroxidase conjugated streptavidin (Cat. 555142, BD Biosciences, USA) were added and incubated at room temperature for 1 hour. The plate was washed again, then 3,3',5,5'-Tetramethylbenzidine (TMB) was added to develop color before the reaction was stopped with 2 N $H_2SO_4$. Optical

density was measured at 450 nM by ELISA reader (Tecan Magellan, Switzerland). The concentration of antibodies detecting recombinant human IFN-γ was calculated from a standard curve. The anti-IFN-γ autoantibody titer was the lowest dilution of plasma that could inhibit recognition between recombinant human IFN-γ and the capture or detection antibodies by 50%.

### Data analysis

All statistical analysis was performed using GraphPad Prism version 6 (GraphPad Software, USA). Continuous variables are reported as median. The comparison among sample group more than three groups was analyzed by One-way ANOVA (Kruskal-Wallis) and post-test using Dunn's Multiple Comparison test and Mann-Whitney test in two groups comparison with non-normal distribution. A receiver operating characteristic (ROC) curve was used to determine cut-off value. Sensitivity, specificity, positive predictive value (PPV) and negative predictive value (NPV) were analyzed with Chi-square (and Fisher's exact) test. Statistically significant difference was determined at P value < 0.05. The statistical power of the study was calculated by post-hoc power analysis for all experiments and with >80% power and 95% confidence to detect differences between groups.

## Results

### General demographic data revealed NTM culture positive disseminated lymphadenopathy patients with anti-IFN-γ autoantibodies and signs of active infection

Unidentified disseminated lymphadenopathy patients had anti-IFN-γ autoantibody detection at the laboratory unit of Srinagarind Hospital, Khon Kaen, Thailand. Positive results for anti-IFN-γ autoantibody were reported in 137 cases, 65 cases had no history of bacterial culture request. Presence of anti-IFN-γ autoantibody in plasma has been recognized as one indicator for disseminated NTM infection diagnosis in adults [8–10]. Only 34 cases were reported with confirmed positive NTM culture, while 26 cases were negative. Surplus plasma samples from routine assay were collected from storage at Srinagarind Hospital, Khon Kaen, while blood samples from tuberculosis patients (MTB, n = 18), melioidosis (BP, n = 19) and healthy controls (HC, n = 30) were collected at Nakhon Panom Hospital, Nakhon Panom, Thailand. Plasma samples were aliquoted, stored at -80˚C and thawed only once at the day of the assay to avoid any freeze-thaw effect.

Comparison of the demographic data from disseminated lymphadenopathy patients with NTM culture positive versus the control groups is shown in Table 1. As expected, only NTM infected patients had anti-IFN-γ autoantibody, it was not present in tuberculosis, melioidosis or healthy control groups. The hematological parameters of NTM patients were similar to those in the melioidosis group, with decreased hemoglobin levels and increased numbers of white blood cells in comparison to healthy controls. Thus, it is likely that the NTM cases in this study had active infections. The most common organ involvement of NTM cases was lymph node (21/34, 61.8%), followed by skin and soft tissue (13/34, 31.2%), bone marrow (2/34, 5.9%), lung (2/34, 5.9%) and blood (2/35, 5.9%). The most common infecting species were *M. abscessus* complex (18/34, 52.9%), followed by *M. avium* complex (MAC) (9/34, 26.5%), *M. scrofulaceum* (2/34, 5.9%), *M. chelonae* (1/34, 2.9%), and *M. szulgai* (1/34, 2.9%). There were 3 cases with unidentified rapid growing mycobacteria (RGM).

**Table 1. Demographic and clinical characteristics of participants.**

| Characteristics | | | NTM | MTB | BP | Healthy | P-value |
|---|---|---|---|---|---|---|---|
| | | | (n = 34) | (n = 18) | (n = 19) | (n = 30) | |
| Age (years), median (range) | | | 53 (31–73) | 47 (18–90) | 57 (19–80) | 44 (21–78) | ns |
| Male, n (%) | | | 18 (53%) | 17 (94%) | 12 (63%) | 14 (47%) | < 0.01 [b,d] |
| Laboratory finding, median (range) | | | | | | | |
| | Anti-IFN-γ autoantibody | | | | | | |
| | | Positive | 34 | 0 | 0 | 0 | N/A |
| | | Negative | 0 | 18 | 19 | 30 | N/A |
| | Hematologic parameter | | | | | | |
| | Hemoglobin (g/dl) | | 10.1 | 10.3 | 8.7 | 14.1 | < 0.0001 [a,c] |
| | | | (6.7–14.5) | (4.8–16.8) | (5.3–12.4) | (6.2–17.4) | < 0.001 [b] |
| | White blood cell count (x 10^6/ml) | | 15.6 | 7.4 | 11.5 | 6.5 | < 0.0001 [a] |
| | | | (5.6–39.2) | (2.7–21.1) | (4.4–32) | (4.3–10.2) | < 0.01 [c,d] |
| | | Neutrophil (x 10^6/ml) | 11.9 | 4.7 | 9.1 | 3.5 | < 0.0001 [a] |
| | | | (3.2–31.1) | (1.0–18.7) | (3.6–26.3) | (1.9–7.9) | < 0.001 [c] |
| | | | | | | | < 0.05 [d] |
| | | Lymphocyte (x 10^6/ml) | 3.1 | 1.4 | 1.6 | 2.2 | < 0.0001 [d] |
| | | | (0.5–10.2) | (0.3–3.6) | (0.5–3.8) | (1.2–4.5) | < 0.01 [e] |
| | | Monocyte (x 10^6/ml) | 0.8 | 0.5 | 0.7 | 0.3 | < 0.0001 [a] |
| | | | (0.3–2.0) | (0.07–3.2) | (0.04–1.9) | (0.1–0.6) | < 0.05 [c] |
| Organ involvement, cases (%) * | | | | | | | |
| | Lymph node | | 21 (61.8%) | - | - | - | N/A |
| | Skin and soft tissue | | 13 (38.2%) | - | 7 (36.8%) | - | N/A |
| | Bone marrow | | 2 (5.9%) | - | 2 (10.5%) | - | N/A |
| | Lung | | 2 (5.9%) | 18 (100%) | - | - | N/A |
| | Blood | | 2 (5.9%) | - | 8 (42.1%) | - | N/A |
| | Liver | | - | - | 2 (10.5%) | - | N/A |
| NTM species, cases (%) | | | | | | | |
| | *M. abscessus* complex | | 18 (52.9%) | - | - | - | N/A |
| | *M. avium* complex | | 9 (26.5%) | - | - | - | N/A |
| | *M. scrofulaceum* | | 2 (5.9%) | - | - | - | N/A |
| | *M. chelonae* | | 1 (2.9%) | - | - | - | N/A |
| | *M. szulgai* | | 1 (2.9%) | - | - | - | N/A |
| | Unidentified RGM species | | 3 (8.8%) | - | - | - | N/A |

NTM; non-tuberculous mycobacteria, MTB; *M. tuberculosis*, BP; *B. pseudomallei*.

*; organ involvements were listed from each culture episode, some patients had multi-organ infection. N/A; not available, -; no report. Significant differences were found between

[a]; NTM vs healthy

[b]; MTB vs healthy

[c]; BP vs healthy

[d]; NTM vs MTB

[e]; NTM vs BP, ns; non significance.

## Anti-GPL-core IgA and IgG antibodies both had high sensitivity, but IgG was more specific than IgA for diagnosis of NTM infection

Plasma samples were evaluated for anti-GPL-core IgA with commercial ELISA kit, while IgG detection was carried out with our modified ELISA protocol. Anti-GPL-core IgA and IgG

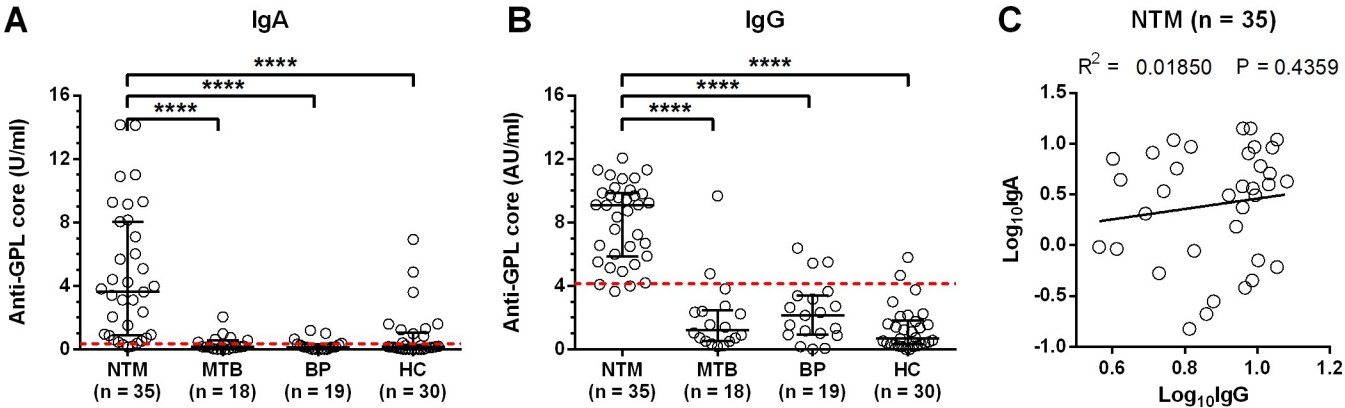

**Fig 1. Detection of anti-GPL core IgA and IgG in plasma samples from subjects with or without NTM infection.** A total of 101 subjects were classified as infected with disseminated non-tuberculosis mycobacteria (NTM, n = 34), infected with *M. tuberculosis* (MTB, n = 18), infected with *B. pseudomallei* (BP, n = 19), and non-infected healthy controls (n = 30). All plasma samples were tested for anti-GPL core IgA with ELISA kit or modified ELISA for detection of anti-GPL core IgG. Results are shown as scatter dot plots for each subject with lines indicating the median and interquartile range. Red dash-line represents the positive cut-off. Anti-GPL core IgA (**A**) or IgG levels (**B**) were compared among different type of infection groups. Statistical differences among the different groups were analyzed by one-way ANOVA (Kruskal-Wallis) and post-test using Dunn's Multiple Comparison test, ****; P < 0.0001. Correlation of anti-GPL core IgA and IgG level from the same NTM infected patients were transformed to $log_{10}$ before linear regression analysis (**C**).

levels from NTM patients were significantly higher than tuberculosis, melioidosis or healthy control groups with P value < 0.0001 (Fig 1A and 1B). But there was no statistical correlation between IgA and IgG levels from NTM patients (Fig 1C).

Cut-off levels for discrimination of NTM patients from other control groups were calculated with a receiver operating characteristic (ROC) curve for both IgA and IgG at the point with sensitivity more than 90% (S2 Fig). Diagnostic efficacy for NTM patients using anti-GPL-core IgA and IgG cut-off at > 0.352 U/ml (area under curve, AUC = 0.8998) and > 4.140 AU/ml (AUC = 0.9723), respectively, are shown in Table 2. Sensitivities for the IgA and IgG cut-offs were the same at 91.18% and the negative predictive values (NPV) were also comparable at 94% and 95.24% for IgA and IgG, respectively. The sensitivity of detection by anti-GPL-core IgG was higher than IgA (IgA at 70.15% and IgG at 89.55%) as was the positive predictive value (PPV; IgA at 60.78% and IgG at 81.58%). Comparisons between NTM patients versus all controls or some control groups (MTB and BP, MTB only, or HC only) also showed higher detection specificities and PPVs for IgG than for IgA (S1 Table).

## Plasma anti-GPL-core IgG levels in patients with SGM were higher than in patients with RGM-NTM

Out of 35 NTM patients, 13 cases were infected with slow growing mycobacteria (SGM) and 22 cases were infected with rapid growing mycobacteria (RGM). Plasma anti-GPL-core IgA levels from either SGM or RGM were not statistically different (Fig 2A). In contrast, plasma anti-GPL-core IgG levels from patients with SGM infection were statistically higher than those for patients with RGM infection (P value = 0.0169, Fig 2B). To address effectiveness of anti-GPL-core IgG to discriminate between SGM and RGM, a cut-off was calculated by using a ROC curve at the point with sensitivity more than 80% and AUC 0.7535 (S3 Fig). At the anti-GPL-core IgG cut off level of more than 7.955 AU/ml, sensitivity to distinguish SGM from RGM was 83.33% with 59.09% specificity, 52.63% PPV and 86.67% NPV (Fig 2C). However, despite the very high sensitivity of the assay, 9 of the 22 patients with RGM (40.9%) had anti-GPL-core IgG levels above the cut-off due to the low specificity of the assay. Interestingly, 11/22 (50%) of the patients with anti-GPL-core IgG levels between the non-NTM and SGM cut-

**Table 2. NTM infection diagnosis efficacies by measuring level of plasma anti-GPL core IgA or IgG.**

| | Positive cut-off value for NTM | Subjects with NTM infection (NTM, n = 34) | | Subjects without NTM infection (MTB + BP + HC, n = 67) | |
| --- | --- | --- | --- | --- | --- |
| | | No. of positive samples / total no. of samples | % sensitivity (95% CI) | No. of negative samples / total no. of samples | % specificity (95% CI) |
| **Anti-GPL core IgA** | > 0.352 | 31/35 | 91.18 | 42/67 | 70.15 |
| | | | (76.32–98.14) | | (57.73–80.72) |
| **Anti-GPL core IgG** | > 4.140 | 31/35 | 91.18 | 60/67 | 89.55 |
| | | | (76.32–98.14) | | (79.65–95.70) |
| | | No. of NTM samples / total no. of positive samples | % PPV (95%CI) | No. of non-NTM samples / total no. of negative samples | % NPV (95%CI) |
| **Anti-GPL core IgA** | > 0.352 | 31/52 | 60.78 | 47/50 | 94.00 |
| | | | (46.11–74.16) | | (83.45–98.75) |
| **Anti-GPL core IgG** | > 4.140 | 31/39 | 81.58 | 60/63 | 95.24 |
| | | | (65.67–92.26) | | (86.71–99.01) |

NTM; non-tuberculous mycobacteria, MTB; *M. tuberculosis*, BP; *B. pseudomallei*, HC; healthy control. PPV; positive predictive value, NPV; negative predictive value, CI; confidence interval.

offs (i.e. 4.140–7.955 AU/ml) were infected with RGM (Fig 2B). Taking all the data together, it is probable that, at moderate levels, the anti-GPL-core IgG not only shows a correlation with NTM but also with RGM, while at high levels it does not distinguish between RGM and SGM.

## Discussion

In this study, we address the diagnostic efficacy of anti-GPL-core ELISA kit in disseminated lymphadenopathy patients positive for NTM culture and anti-IFN-γ autoantibodies. Both anti-GPL-core IgA and IgG levels were significantly higher in NTM patients than tuberculosis, melioidosis or healthy control groups. Using a cut-off value of 4.140 AU/ml for anti-GPL-core IgG provided good sensitivity for identifying the NTM patients with increased specificity and %PPV than anti-GPL-core IgA. Furthermore, most patients with moderate levels of anti-GPL-core IgG could be identified to have RGM infection.

According to our sample collection, enlarged lymph nodes were the most common NTM organ involvement followed by skin and soft-tissue infection with a minority of patients infected at other sites. Half of the NTM species in our patient collection were *M. abscessus* followed by MAC and other species of rapid and slow growing mycobacteria. This observation is consistent with previous publications on NTM infection in Thai and Chinese patients [4, 9, 10, 21, 25]. However, the infecting species and the sites of infection are different from those in Thailand in some regions, including Japan, Taiwan and the USA [25]. The epidemiology of NTM in these regions is reported as higher numbers of pulmonary NTM compared with extrapulmonary NTM. Moreover, MAC is the most common species found in these patients [26–30]. Due to the pathogenic NTM are geometrically various, therefore, this study may be limited in Thailand and/or *M. abcessus*-dominant country such as Indonesia, southern China, and/or Taiwan.

The anti-GPL-core IgA ELISA kit has been developed for diagnosis of MAC pulmonary disease on the basis of that GPL-core is a major antigen on cell wall of several *Mycobacterium* species, except MTB complex and *M. kansasii* [17]. This diagnostic kit has been previously evaluated in pulmonary NTM infection patients in Japan, Taiwan, South Korea and USA [17, 19, 31–34]. Studies in Japanese pulmonary infected patients reported that the sensitivity for diagnosis of MAC disease was more than 90% with specificity at more than 95% [17, 31].

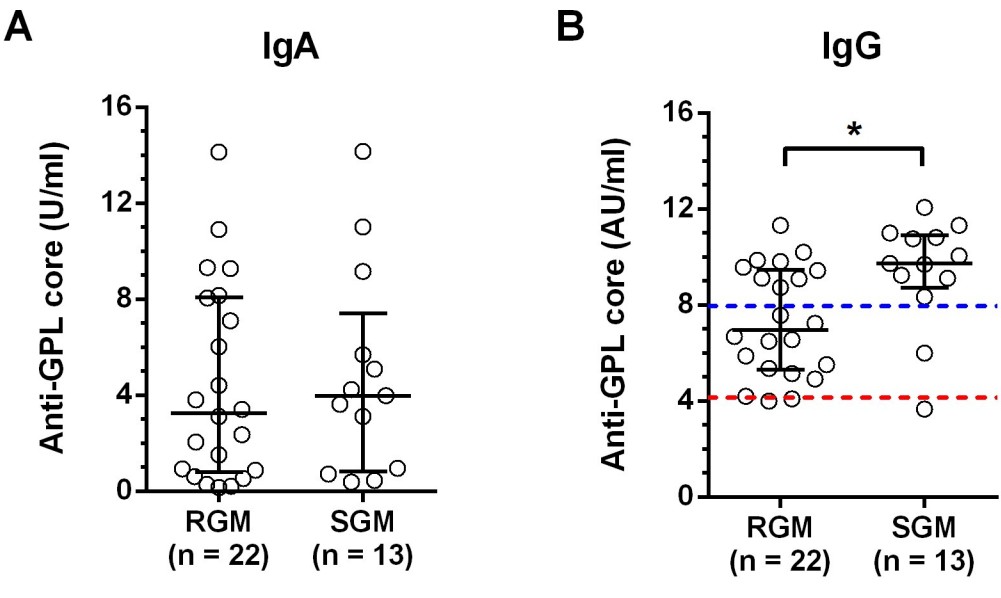

| | Positive cut-off value for SGM | Subjects with SGM infection (SGM, n = 13) | | Subjects with RGM infection (RGM, n = 22) | |
|---|---|---|---|---|---|
| | | No. of SGM samples / total no. of samples | % sensitivity (95%CI) | No. of RGM samples / total no. of samples | % specificity (95%CI) |
| Anti-GPL core IgG | > 7.955 | 11/13 | 84.62 (54.55 – 98.08) | 13/22 | 59.09 (36.35 – 79.29) |
| | Positive cut-off value for SGM | NTM subjects with anti-GPL core IgG positive (n = 20) | | NTM subjects with anti-GPL core IgG negative (n = 15) | |
| | | No. of SGM samples / total no. of positive samples | % PPV (95%CI) | No. of RGM samples / total no. of negative samples | % NPV (95%CI) |
| Anti-GPL core IgG | > 7.955 | 11/20 | 55.00 (31.53 – 76.94) | 13/15 | 86.67 (59.54 – 98.34) |

PPV; positive predictive value, NPV; negative predictive value

**Fig 2. Discrimination between Slow Growing Mycobacteria (SGM) or Rapid Growing Mycobacteria (RGM) by quantification of anti-GPL core IgG.** Plasma anti-GPL core IgA (**A**) or IgG level (**B**) from patients with NTM culture positive confirmed as SGM (n = 13) or RGM (n = 22). Results are shown as scatter dot plot from each subject with lines indicating the median and interquartile range. Red dash-line represents the NTM positive cut-off while blue dash-line represents the SGM positive cut-off. Statistical differences between infected patients with RGM versus SGM were analyzed with Mann-Whitney U test, *; P < 0.05.

Another study in South Korea reported the sensitivity for diagnosis of NTM (MAC and non-MAC) disease ranged from 77.5–85%, with specificity at 100% [32]. Interestingly, when discriminating between MAC disease and MAC colonization, a study in Taiwan reported lower

sensitivity at 60% and specificity at 91% [34], and a study in American pulmonary infection patients showed 70.1% sensitivity and 93.9% specificity [33]. In our study, the diagnostic efficacy for disseminated NTM disease by using anti-GPL-core IgA showed a good sensitivity at 91.18% but with an intermediate specificity at 70.15%. The intermediate specificity we observed might be explained by differences in the clinical manifestations of our patients. Pulmonary infections usually induce high levels of IgA antibody with systemic infections usually IgG dominant [35]. Supported by our finding that plasma anti-GPL-core IgA and IgG were not significantly correlated, this means that the magnitude of the plasma IgA level could not differentiate between disseminated NTM infection patients and subjects without NTM infection. A meta-analysis study in the diagnostic accuracy of anti-GPL core IgA in MAC pulmonary diseases (MAC-PD) indicated 69.6% (95% CI 62.1–76.1%) sensitivity and 90.6% (95% CI 83.6–95.1) specificity [20]. Our analysis of anti-GPL core IgA for disseminated NTM cases resulted in 90% specificity and the sensitivity was 68.6% with the cut-off > 1.264 U/ml (S2 Fig). This shows that the anti-GPL core IgA accuracy is consistent in pulmonary MAC and disseminated NTM diseases.

Detection of anti-GPL-core IgG instead of IgA was applied in our study for the diagnosis of disseminated NTM patients. Our results show that anti-GPL-core IgG and IgA had the same sensitivity at 91.18% with comparable NPVs at 95.24% and 94% for IgG and IgA, respectively. In more details, our data showed that anti-GPL-core IgG had some reactivities in MTB, BP patients and healthy controls which could be two possibilities; 1) plasma IgG against NTM could be pre-existing in people living in the area due to exposure to environmental NTM [36, 37] and/or 2) interference signals due to polyreactive IgG since ELISA detected every IgG including specific and non-specific or polyreactive antibodies [38]. As expected, the detection of anti-GPL-core IgG increased the test specificity to 89.55% and PPV to 81.58% compared with IgA detection. On the other hand, a previous study using the same anti-GPL-core ELISA kit for detection of IgG for diagnosis of MAC pulmonary disease showed low sensitivity (72.6%) but good specificity at 92.2% [31]. According to the symptoms of disseminated NTM infection which are not specific, diagnosis of the patients requires a high degree of suspicion and can be improved by measuring serum antibodies [39]. Our data support the idea that targeting anti-GPL-core assay, especially IgG, is suitable for supportive diagnosis of disseminated NTM patients along with detection of anti-IFN-γ autoantibodies due to the lack of sensitivity of NTM culture from clinical specimens. The other studies of anti-GPL-core IgA level from MAC-PD represents disease activity which is useful for monitoring of the disease progression [15, 40, 41]. Further studies of anti-GPL-core IgA and IgG in association with clinical progression of disseminated NTM infection should be investigated.

There was no difference in anti-GPL-core IgA levels between SGM and RGM, but anti-GPL-core IgG levels from patients with SGM infection were significantly higher than for the RGM infection group. This observation is similar to a study conducted in South Korean patients that found MAC-PD was not able to be distinguished from *M. abscessus* complex pulmonary disease (MAB-PD) by using anti-GPL-core IgA ELISA [32]. According to the different levels of anti-GPL-core IgG between the SGM and RGM groups in our study, most patients who had an intermediate level of anti-GPL-core IgG (4.140–7.955 AU/ml) could be identified as RGM infection (11/12, 92%). Thus, this study demonstrates the capacity of anti-GPL-core antibody to potentially discriminate between SGM and RGM.

In conclusion, this study is the first evaluation of a commercially available ELISA kit for detection of anti-GPL core IgA in disseminated NTM infection patients with anti-IFN-γ autoantibodies. Our data indicate that determination of either anti-GPL-core IgA or IgG level is suitable for these patients with good sensitivity at 91.43%; however, IgG level was more specific (89.55%) and had a higher PPV (81.58%). Disseminated NTM diseases is hard to come to a

precise diagnosis so availability of this kit for supportive diagnosis of NTM disease could make us possible to approach a precise diagnosis of them. This study also provides an interesting insight into discrimination between SGM and RGM infection. High levels of IgG (more than 7.955 AU/ml) can be identified as either SGM or RGM, but patients with moderate IgG level (4.14–7.955 AU/ml) are likely to have RGM infection. Taken together, the application of anti-GPL-core antibody detection provides a novel additional option for clinicians to diagnose and manage disseminated NTM infected patients.

## Supporting information

**S1 Fig. Human plasma anti-GPL-core IgG standard curve analyzed by linear regression.** Human reference plasma was diluted at 10, 7.5, 5, 2.5, and 1.25 AU/ml. Optical density (O.D.) of each concentration was plotted before analyzing with linear regression by using Prism GraphPad Software.
(DOCX)

**S2 Fig. Determination of positive cut-off for NTM diagnosis by detecting level of plasma anti-GPL core IgA or IgG.** Receiver operating characteristic (ROC) curve analysis was applied to determine area under curve (AUC) and positive cut-off for NTM infection diagnosis by detecting anti-GPL core IgA **(A)** or IgG level **(B)**. Red arrow represents cut-off point on ROC curve at 90% sensitivity. Blue arrow represents cut-off point on ROC curve at 90% specificity.
(DOCX)

**S3 Fig. Determination of positive cut-off to discriminate Slow Growing Mycobacteria (SGM) from Rapid Growing Mycobacteria (RGM) by quantification of anti-GPL core IgG.** Receiver operating characteristic (ROC) curve analysis was applied to determine area under curve (AUC) and positive cut-off for SGM infection prediction by detecting anti-GPL core IgG level. Red arrow represents cut-off point on ROC curve.
(DOCX)

**S1 Table. Specificity, PPV and NPV of positive cut-off value to distinguish NTM infection patients from different group of subjects without NTM infection.**
(DOCX)

## Acknowledgments

The authors thank Ms. Jeerawan Dhanasen and Ms. Anchalee Tiyabut for assistance with patient hospital record review and Dr. Glenn Borlace for English-language assistance under the aegis of the Publication Clinic, Khon Kaen University.

## Author Contributions

**Conceptualization:** Ploenchan Chetchotisakd, Manabu Ato, Ganjana Lertmemongkolchai.

**Data curation:** Arnone Nithichanon, Waraporn Samer.

**Formal analysis:** Arnone Nithichanon, Ploenchan Chetchotisakd, Chidchamai Kewcharoen-wong, Manabu Ato, Ganjana Lertmemongkolchai.

**Funding acquisition:** Manabu Ato, Ganjana Lertmemongkolchai.

**Investigation:** Waraporn Samer, Chidchamai Kewcharoenwong.

**Methodology:** Arnone Nithichanon, Waraporn Samer.

**Supervision:** Chidchamai Kewcharoenwong, Ganjana Lertmemongkolchai.

**Validation:** Arnone Nithichanon, Ploenchan Chetchotisakd.

**Writing – original draft:** Arnone Nithichanon, Waraporn Samer.

**Writing – review & editing:** Ploenchan Chetchotisakd, Chidchamai Kewcharoenwong, Manabu Ato, Ganjana Lertmemongkolchai.

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
