## [Decision Letter · Decision Letter 0]

19 Aug 2020

PONE-D-20-21272

Evaluation of plasma anti-GPL-core IgA and IgG for Diagnosis of Disseminated Non-tuberculous Mycobacteria Infection

PLOS ONE

Dear Dr. Lertmemongkolchai,

Thank you for submitting your manuscript to PLOS ONE. After careful consideration, we feel that it has merit but does not fully meet PLOS ONE’s publication criteria as it currently stands. Therefore, we invite you to submit a revised version of the manuscript that addresses the points raised during the review process.

Please submit your revised manuscript. If you will need more time than this to complete your revisions, please reply to this message or contact the journal office at plosone@plos.org. Please include the following items when submitting your revised manuscript:

We look forward to receiving your revised manuscript.

Kind regards,

Frederick Quinn

Academic Editor

PLOS ONE

Journal Requirements:

2. Please provide additional details regarding participant consent. In the ethics statement in the Methods and online submission information, please clarify what type of consent you obtained (for instance, written or verbal, and if verbal, how it was documented and witnessed). If your study included minors, state whether you obtained consent from parents or guardians.

3. Please provide the catalog numbers for the anti-GPL core IgA ELISA kit and all antibodies used in the study.

 "This study was supported in part by the Japan Agency for Medical Research and Development (AMED) (Grant number: JP20fk0108129, JP20fk0108139) to MA. The funder had no role in study design, data collection and analysis, decision to publish, or preparation of the manuscript."

Reviewers' comments:

Reviewer's Responses to Questions

**Comments to the Author**

1. Is the manuscript technically sound, and do the data support the conclusions?

Reviewer #1: Yes

Reviewer #2: Partly

2. Has the statistical analysis been performed appropriately and rigorously? 

Reviewer #1: Yes

Reviewer #2: Yes

3. Have the authors made all data underlying the findings in their manuscript fully available?

Reviewer #1: Yes

Reviewer #2: Yes

4. Is the manuscript presented in an intelligible fashion and written in standard English?

Reviewer #1: Yes

Reviewer #2: Yes

5. Review Comments to the Author

Reviewer #1: Manuscript Review: PONE-D-20-21272

Evaluation of plasma anti-GPL-core 1 IgA and IgG for Diagnosis of Disseminated Non-tuberculous Mycobacteria Infection

Arnone Nithichanon, Waraporn Samer, Ploenchan Chetchotisakd, Chidchamai

Kewcharoenwong, Manabu Ato, Ganjana Lertmemonkolchai

Key Results:

In the manuscript the authors attempt to determine the efficacy of an anti-GPL-core ELISA kit, originally developed for detection of Mycobacterium avium species, in disseminated lymphadenopathy patients positive for non-tuberculous Mycobacteria (NTM) and anti-interferon gamma autoantibodies. Since the typical diagnosis of NTM does not have good sensitivity and is time consuming, this technique would improve the detection of NTM in patients in this region of the world. 137 plasma samples from a hospital in Khon Kaen, Thailand were analyzed along with control blood samples positive for M. tuberculosis, B. pseudomallei or healthy individuals. The study determined that the detection sensitivity of NTM was approximately 91% and specificity was 70%. Of those samples analyzed, results showed that only patients infected with NTM were positive for anti-IFN gamma autoantibody. Furthermore, anti-GPL-core IgA and IgG levels of NTM patients, including test sensitivities, were also higher than the control groups. When trying to determine if the test was able to distinguish between slow or rapid growing Mycobacteria, the study found that it was only able to distinguish between the two when immunoglobulin levels were at moderate levels; high levels showed no distinction. Overall, the study determined that using the kit does indeed improve the detection of NTM over current methods.

Validity:

Based on the methodology used in this study, I find the manuscript to be valid.

Originality and Significance:

I find the data presented in this study to be original and significant. If this study were to be expanded to other regions of the world and have the same results, it may shift the way testing for the infection occurs.

Data and Methodology:

This manuscript presented a straightforward data analysis based on data collected through analysis of blood samples. The methodology presented is commonly used and appropriate.

Appropriate us of Statistics:

Appropriate statistics were used throughout the study. Proper statistics were conducted, and variables were adjusted for when needed.

Conclusions:

Based on the statistics presented in the study, the conclusions appear to be valid and reliable.

Suggested Improvements:

None

References:

The references are valid.

Clarity and context:

The abstract, introduction and conclusions are clear, concise and appropriate.

Scope of expertise:

This manuscript is within the scope of my expertise.

Reviewer #2: The aim of the manuscript was to evaluate surplus plasma samples from Thai patients suspected of disseminated NTM with an anti-GPL-core IgA kit and the modified kit to also detect IgG to compare the diagnostic potential of both isotypes.

The authors' approach is interesting, however serology IgA-GPL has some limitation, that is import for clinicians. As described by the authors and others GLP is present in MAC, RGM, including M. abscessus, and IgA-GPL core ELISA cross-reactivity between MAC and RGM in pulmonary cases is reported. It is well known that the drug treatment regime is different for MAC and M. abscessus. However, both IgA and IgG tests failed to distinguish them in disseminated NTM cases. Even significantly higher IgG median reactivity was described for RGM the test weakness remains because the number of tested samples is too small.

According to the reference cited by the authors, a meta-analysis study in the diagnostic accuracy of IgA-GPL core in MAC pulmonary diseases was 0.696 (95% CI 0.621–0.761) and 0.906 (95% CI 0.836–0.951), for sensitivity and specificity, respectively. Looking at the results of the present study the IgA sensitivity (61.7) and specificity (95) are similar if use a higher cut off for disseminated NTM cases. This shows the IgA-GPL core accuracy is consistent in pulmonary MAC and disseminated NTM diseases. This discussion is missed.

On the other hand, the IgG test showed higher reactivity distribution in disseminated NTM samples, but the other infections disease and control also showed dispersion and cross-reactivity. As the number of MTB and BP samples is too small, thus may cause bias in the test specificity. How do the authors explain this cross-reactivity?

All this makes the authors' assumption “Taken together, the application of anti-GPL-core antibody detection provides a novel additional option for clinicians to diagnose and manage disseminated NTM infected patients” speculative. The authors should discuss what can be the role of the biomarkers-based assays for the diagnosis of disseminated NTM and treatment management considering the limitations of the test.

The major limitation of the study is the sample size of MTB and BP cases. With few subjects included is difficult to really assess the role of the IgG test, although for IgA the accuracy was consistent for both MAC pulmonary disease and disseminated NTM.

Other comments:

Line 104-104 - Consider to change: To detect anti-GPL-core IgG, the antigen adsorbed plate of the IgA kit was used to perform the modified test. Briefly, .....

Line 107 - Biotinylated ....anti-human IgG (BD Biosciences, USA). It was made in what animal?

Line 112-113 - ... Results were analyzed with in-house standard curve and reported as arbitrary unit/ml (AU/ml). How did you that? Did you use the software? Manual mathematics calculation? Which plasma reference was used?

Line 128 – 129 - --- biotinylated anti-human IFN-γ (BD Biosciences, USA). The conjugate was titrated?

Seems that it was used surplus plasma samples of suspected disseminated NTM cases from a Hospital bio-repository and the control was freshly collected. However, the authors did not comment on the storage condition of the samples. This is an important issue in serology because Ab degradation can alter the ELISA results.

Line 159 - .....plasma samples from routine assay were collected from storage at..... Inform the storage condiction (temperature? suffered freeze-thaw?)

6. PLOS authors have the option to publish the peer review history of their article (what does this mean?). If published, this will include your full peer review and any attached files.

Reviewer #1: No

Reviewer #2: No

---

## [Author Response · Author response to Decision Letter 0]

11 Oct 2020

Manuscript Review: PONE-D-20-21272

Title: Evaluation of plasma anti-GPL-core 1 IgA and IgG for Diagnosis of Disseminated Non-tuberculous Mycobacteria Infection

Authors: Arnone Nithichanon, Waraporn Samer, Ploenchan Chetchotisakd, Chidchamai Kewcharoenwong, Manabu Ato, Ganjana Lertmemonkolchai

Reviewer #1:

Key Results: Overall, the study determined that using the kit does indeed improve the detection of NTM over current methods.

Validity: Based on the methodology used in this study, I find the manuscript to be valid.

Originality and Significance: I find the data presented in this study to be original and significant. If this study were to be expanded to other regions of the world and have the same results, it may shift the way testing for the infection occurs.

Data and Methodology: This manuscript presented a straightforward data analysis based on data collected through analysis of blood samples. The methodology presented is commonly used and appropriate.

Appropriate use of Statistics: Appropriate statistics were used throughout the study. Proper statistics were conducted, and variables were adjusted for when needed.

Conclusions: Based on the statistics presented in the study, the conclusions appear to be valid and reliable.

Suggested Improvements: None

References: The references are valid.

Clarity and context: The abstract, introduction and conclusions are clear, concise and appropriate.

Scope of expertise: This manuscript is within the scope of my expertise.

The Authors: We are incredibly grateful and mostly appreciate the positive assessments.

Reviewer #2: 

The aim of the manuscript was to evaluate surplus plasma samples from Thai patients suspected of disseminated NTM with an anti-GPL-core IgA kit and the modified kit to also detect IgG to compare the diagnostic potential of both isotypes.

The authors' approach is interesting, however serology IgA-GPL has some limitation, that is import for clinicians. As described by the authors and others GLP is present in MAC, RGM, including M. abscessus, and IgA-GPL core ELISA cross-reactivity between MAC and RGM in pulmonary cases is reported. It is well known that the drug treatment regime is different for MAC and M. abscessus. However, both IgA and IgG tests failed to distinguish them in disseminated NTM cases. Even significantly higher IgG median reactivity was described for RGM the test weakness remains because the number of tested samples is too small.

According to the reference cited by the authors, a meta-analysis study in the diagnostic accuracy of IgA-GPL core in MAC pulmonary diseases was 0.696 (95% CI 0.621–0.761) and 0.906 (95% CI 0.836–0.951), for sensitivity and specificity, respectively. Looking at the results of the present study the IgA sensitivity (61.7) and specificity (95) are similar if use a higher cut off for disseminated NTM cases. This shows the IgA-GPL core accuracy is consistent in pulmonary MAC and disseminated NTM diseases. This discussion is missed.

The Authors: We mostly appreciate the comments and have revised the manuscript as kindly suggested: Line 293 – 299 “A meta-analysis study in the diagnostic accuracy of anti-GPL core IgA in MAC pulmonary diseases (MAC-PD) indicated 69.6% (95% CI 62.1 – 76.1%) sensitivity and 90.6% (95% CI 83.6 – 95.1) specificity [20]. Our analysis of anti-GPL core IgA for disseminated NTM cases resulted in 90% specificity and the sensitivity was 68.6% at the cut-off > 1.264 U/ml (Fig S2). This shows that the anti-GPL core IgA accuracy is consistent in pulmonary MAC and disseminated NTM diseases.” Additionally, we also modified the Supplementary Figure S2 to support the discussion as below.

 Supplementary Figure S2: Determination of positive cut-off for NTM diagnosis by detecting level of plasma anti-GPL core IgA or IgG. Receiver operating characteristic (ROC) curve analysis was applied to determine area under curve (AUC) and positive cut-off for NTM infection diagnosis by detecting anti-GPL core IgA (A) or IgG level (B). Red arrow represents cut-off point on ROC curve at 90% sensitivity. Blue arrow represents cut-off point on ROC curve at 90% specificity.

On the other hand, the IgG test showed higher reactivity distribution in disseminated NTM samples, but the other infections disease and control also showed dispersion and cross-reactivity. As the number of MTB and BP samples is too small, thus may cause bias in the test specificity. How do the authors explain this cross-reactivity? 

The Authors: We added more discussion that the other infectious diseases and control also showed dispersion and cross-reactivity IgG as following: Line 303 – 308 “In more details, our data showed that anti-GPL-core IgG had some reactivity in MTB, BP patients and healthy controls which could be explained with two possibilities; 1) plasma IgG against NTM could be pre-existing in people in the area due to exposure to NTM in environment [36, 37] and/or 2) interference signals due to polyreactive IgG since ELISA detected every IgG including specific and non-specific or polyreactive antibodies [38].”

Regarding the comment that the number of MTB and BP samples may be too small. We analyzed the power of statistical comparison by post-hoc analysis and it resulted in 100% power, as shown below. In addition, the data in Supplementary Table S1: % specificity of the assay to distinguish disseminated NTM patients from all controls (MTB + BP + HC), MTB+BP, MTB, or healthy controls show 89.55%, 86.49%, 88.89% and 93.33%, respectively. These % specificities are remarkably close to each other. Therefore, the bias in the study could be minimal. 

All this makes the authors' assumption “Taken together, the application of anti-GPL-core antibody detection provides a novel additional option for clinicians to diagnose and manage disseminated NTM infected patients” speculative. The authors should discuss what can be the role of the biomarkers-based assays for the diagnosis of disseminated NTM and treatment management considering the limitations of the test. 

The Authors: We modified the discussion as suggested: Line 311 – 320 “According to the symptoms of disseminated NTM infection which are not specific, diagnosis of the patients requires a high degree of suspicion and can be improved by measuring serum antibodies [39]. Our data support the idea that targeting anti-GPL-core assay, especially IgG, is suitable for supportive diagnosis of disseminated NTM patients along with detection of anti-IFN-� autoantibodies due to the lack of sensitivity of NTM culture from clinical specimens. The other studies of anti-GPL-core IgA level in MAC-PD represents the disease activity which is useful for monitoring of the disease progression [15, 40, 41]. More studies of anti-GPL-core IgA and IgG in association with clinical progression of disseminated NTM infection should be further investigated.”. 

The major limitation of the study is the sample size of MTB and BP cases. With few subjects included is difficult to really assess the role of the IgG test, although for IgA the accuracy was consistent for both MAC pulmonary disease and disseminated NTM.

The Authors: These issues are discussed above.

Other comments:

Line 104-104 - Consider to change: To detect anti-GPL-core IgG, the antigen adsorbed plate of the IgA kit was used to perform the modified test. Briefly, .....

The Authors: We modified the text as suggested: Line 104 – 105, “To detect anti-GPL-core IgG, the antigen adsorbed plate of the IgA kit was used to perform the modified test.”

Line 107 - Biotinylated ....anti-human IgG (BD Biosciences, USA). It was made in what animal?

The Authors: The monoclonal antibody was made in mice. The information of this antibody was clarified, Line 107 and 133 “biotinylated mouse anti-human IFN-γ (Cat. 555142, BD Biosciences, USA)”.

Line 112-113 - ... Results were analyzed with in-house standard curve and reported as arbitrary unit/ml (AU/ml). How did you that? Did you use the software? Manual mathematics calculation? Which plasma reference was used?

The Authors: We added the information: Line 113 – 117 “Results were analysed with in-house standard curve selected from a reference sample, concentrations 10, 7.5, 5, 2.5 and 1.25 Arbitrary unit/ml (AU/ml) and the unknow samples were interpolated from the standard curve by linear regression using GraphPad Prism (Fig S1). Any samples with anti-GPL-core IgG more than 10 AU/ml were diluted and repeated the assay.” Additionally, we also inserted the new Supplementary Figure S1 as below.

Supplementary Figure S1: Human plasma anti-GPL-core IgG standard curve analyzed by linear regression. Human reference plasma was diluted at 10, 7.5, 5, 2.5, and 1.25 AU/ml. Optical density (O.D.) of each concentration was plotted before analyzing with linear regression by using Prism GraphPad Software.

Line 128 – 129 - --- biotinylated anti-human IFN-γ (BD Biosciences, USA). The conjugate was titrated?

The Authors: To quantify level of unbound human IFN-γ, we followed the protocol from manufacturer. The suggested dilution of each reagent was added: Line 121 “1:250 diluted human IFN-γ capture antibody (Cat. 555142, BD Biosciences, USA)”, and Line 133 – 134 “1:250 diluted biotinylated mouse anti-human IFN-γ (Cat. 555142, BD Biosciences, USA) and 1:250 diluted horseradish peroxidase conjugated streptavidin (Cat. 555142, BD Biosciences, USA)”.

Seems that it was used surplus plasma samples of suspected disseminated NTM cases from a Hospital bio-repository and the control was freshly collected. However, the authors did not comment on the storage condition of the samples. This is an important issue in serology because Ab degradation can alter the ELISA results.

The Authors: We declared a statement of plasma sample storage: Line 167 – 168 “Plasma samples were aliquoted, stored at -80oC and thawed only once at the day of the assay to avoid any freeze-thaw effect.”.

Line 159 - .....plasma samples from routine assay were collected from storage at..... Inform the storage condiction (temperature? suffered freeze-thaw?)

The Authors: As above.

---

## [Decision Letter · Decision Letter 1]

6 Nov 2020

Evaluation of plasma anti-GPL-core IgA and IgG for Diagnosis of Disseminated Non-tuberculous Mycobacteria Infection

PONE-D-20-21272R1

Dear Dr. Lertmemongkolchai,

We’re pleased to inform you that your manuscript has been judged scientifically suitable for publication and will be formally accepted for publication once it meets all outstanding technical requirements.

Kind regards,

Frederick Quinn

Academic Editor

PLOS ONE

Additional Editor Comments (optional):

Reviewers' comments:

Reviewer's Responses to Questions

**Comments to the Author**

1. If the authors have adequately addressed your comments raised in a previous round of review and you feel that this manuscript is now acceptable for publication, you may indicate that here to bypass the “Comments to the Author” section, enter your conflict of interest statement in the “Confidential to Editor” section, and submit your "Accept" recommendation.

Reviewer #1: All comments have been addressed

Reviewer #2: All comments have been addressed

2. Is the manuscript technically sound, and do the data support the conclusions?

Reviewer #1: Yes

Reviewer #2: Yes

3. Has the statistical analysis been performed appropriately and rigorously? 

Reviewer #1: Yes

Reviewer #2: Yes

4. Have the authors made all data underlying the findings in their manuscript fully available?

Reviewer #1: Yes

Reviewer #2: Yes

5. Is the manuscript presented in an intelligible fashion and written in standard English?

Reviewer #1: Yes

Reviewer #2: Yes

6. Review Comments to the Author

Reviewer #1: The authors addressed all of my comments in the previous version. I have no additional comments to add.

Reviewer #2: The authors answered the questions clearly and appropriately. Rewrote parts of the original text and improved the discussion

7. PLOS authors have the option to publish the peer review history of their article (what does this mean?). If published, this will include your full peer review and any attached files.

Reviewer #1: No

Reviewer #2: No

---

## [Editor Report · Acceptance letter]

17 Nov 2020

PONE-D-20-21272R1 

Evaluation of plasma anti-GPL-core IgA and IgG for Diagnosis of Disseminated Non-tuberculous Mycobacteria Infection 

Dear Dr. Lertmemongkolchai:

I'm pleased to inform you that your manuscript has been deemed suitable for publication in PLOS ONE. Congratulations! Your manuscript is now with our production department. 

Kind regards, 

on behalf of

Dr. Frederick Quinn 

Academic Editor

PLOS ONE